# Fracture Toughness Testing of Brittle Laminated Geomaterials Using Hollow Double-Wing Slotted Specimens

**DOI:** 10.3390/ma16206754

**Published:** 2023-10-18

**Authors:** Zilong Yao, Zidong Fan, Qin Zhou, Xiaofang Nie, Li Ren

**Affiliations:** 1Key Laboratory of Deep Earth Science and Engineering (MOE), Sichuan University, Chengdu 610065, China; m17373147851_1@163.com (Z.Y.); zq937766477@outlook.com (Q.Z.); xfn@stu.scu.edu.cn (X.N.); renli@scu.edu.cn (L.R.); 2College of Architecture and Environment, Sichuan University, Chengdu 610065, China

**Keywords:** hollow double-wing slotted specimen, fracture toughness, shape factor, numerical calibration

## Abstract

The fracture toughness of shale is a key parameter guiding hydraulic fracturing design and optimization. The hollow double-wing slotted (HDWS) specimen is a typical specimen configuration for measuring the mode I fracture toughness of rock. The calibration of the shape factor (*f*) is the basis for accurately obtaining the fracture toughness of rocks. In this study, the influences of crack length, hole size, and the anisotropy of elastic parameters on f for specimens with three typical bedding orientations—arrester (A), divider (D), and short-transverse (ST) orientations—are systematically investigated using finite element software. The numerical simulation results support the following findings. The mode I *f* increases monotonically with an increase in hole size. The influence of crack length on *f* varies depending on hole sizes. Under different bedding orientations, significant anisotropy in f was observed. In addition, the degree of anisotropy in Young’s modulus has a major impact on *f*, which is related to the bedding orientation of the specimen. The apparent shear modulus ratio has relatively little influence on *f*. As the hole size and crack length increase, the influence of the anisotropy of elastic parameters on f increases. Based on numerical calculations, hydraulic fracturing experiments were conducted on HDWS specimens of Longmaxi shale with three bedding orientations, and the results showed that the peak pressure and fracture toughness of the samples in the ST direction were the lowest, while those in the A direction were the highest.

## 1. Introduction

Natural gas is a clean and low-carbon fossil energy that will play an important role in the global green and low-carbon energy transformation. Shale gas is a type of unconventional natural gas that will play an important role in the growth of China’s natural gas production in the future. Shale gas reservoirs have a very low permeability and no natural production capacity. To develop shale gas, oil and gas reservoirs must be stimulated [1]. Hydraulic fracturing is a common means of reservoir stimulation [2,3]. In hydraulic fracturing stimulation, mode I fractures are one of the most common crack propagation types [4]. As a key parameter that describes the crack propagation resistance of a material, fracture toughness is a core parameter in fracturing design when using classic hydraulic fracturing models such as KGD, PKN, and P3D [5]. Accurately obtaining the mode I fracture toughness of shale is important when designing and optimizing hydraulic fracturing projects.

In the field of rock fracture mechanics, scholars have proposed many specimen configurations and test methods for testing the fracture toughness of rock. For the mode I fracture toughness test, the International Society of Rock Mechanics (ISRM) recommended a cracked chevron-notched Brazilian disc (CCNBD) specimen in 1995 [6] and a semicircular bend (SCB) specimen in 2014 [7]. In addition, the cracked-straight-through Brazilian disc (CSTBD) specimen [8], the cracked-straight-through flattened Brazilian disc (CSTFBD) specimen [9], and the notched deep beam (NDB) specimen [10] are also commonly used in mode I fracture toughness tests. At present, the proposed specimen configurations are mostly suitable for fracture toughness tests performed under normal temperature and pressure conditions. The mechanical properties of rock under deep in situ conditions differ significantly from those under normal temperature and pressure [11,12]. The burial depth of shale gas reservoirs in China generally exceeds 2000 m [13,14], and in situ stress can significantly affect crack propagation in shale gas reservoirs [15,16], which in turn affects the stimulation of shale gas reservoirs. Therefore, it is important to test the fracture toughness of shale under in situ stress to determine the stimulation process of deep shale gas reservoirs [17]. Researchers have carried out measurements and studied rock fracture toughness under simulated in situ stress conditions. For example, Fuentealba et al. [18] used a hydraulic system to pressurize a round bar straight-notched (RBSN) specimen in a sealing system to simulate rock fracture under reservoir in situ stress conditions and experimentally measured the fracture toughness. Using the SCB test method, Kataoka et al. [19] estimated the mode I fracture toughness of rock under confining pressures of 0 to 10 MPa by inserting the specimen into an oil-pressure chamber. Al-Shayea et al. [20] investigated the fracture toughness of rock under a maximum confining pressure of 28 MPa by placing a straight-notched Brazilian disc (SNBD) specimen into an oil pressure chamber and hydraulically increasing the pressure. Zhou Qin et al. [15] investigated the fracture of shale with in situ stresses of 0 MPa, 20 MPa, 40 MPa, and 60 MPa at different lamination angles. The breakdown pressure, mode I fracture toughness, and fracture energy were found to increase significantly with the confining pressure, indicating that the fracture resistance of the studied shale increased remarkably under high in situ stresses. The above results showed that the confining pressure influences the fracture toughness of rock; the higher the confining pressure is, the higher the fracture toughness. A hollow double-wing slotted (HDWS) specimen has a cylindrical profile that facilitates the simulation of in situ stress through the application of confining pressure and can be used to test the fracture toughness of rock subjected to hydraulic fracturing. Therefore, HDWS specimens are often used to test the fracture toughness of rock under different simulated in situ stress conditions.

With other experimental conditions remaining constant, the fracture parameters tested using specimens with different configurations vary, mainly because different specimen configurations have different shape factors (*f*s) (also known as dimensionless stress intensity factors or SIFs). Accurately calibrating *f* for a specimen configuration is a prerequisite for accurately obtaining the fracture toughness. Currently, many researchers have carried out fracture parameter research and *f* calibration for different specimen configurations. Luo et al. [10] tested the rock fracture toughness of a single-edge-notched deep beam (SENDB) specimen under pure mode I, pure mode II, and mixed-mode I/II loading, considering different crack angles and different crack lengths, and they systematically studied and comprehensively and numerically calibrated the dimensionless factors corresponding to three specimen parameters: *K*_I_, *K*_II_, and a novel experimental procedure used by *T*. Jiale Li et al. [21], whereby SCB tests were combined with small-scale direct tensile tests to determine *K*_Ic_ and fracture process zone (FPZ) lengths. The authors performed size effect normalization analysis and found that different types of rock materials satisfy the generalized size effect law to some extent and that testing larger specimens with a relatively small FPZ gives a more objective fracture toughness result. Iqbal et al. [22] experimentally calibrated two fracture toughness test methods, i.e., the chevron bend (CB) test and the CCNBD test, and considered different geometries and materials to verify the effectiveness of the two test methods. Khan et al. [23] calibrated the *f* values for fracture specimens by conducting the straight-edge-cracked round bar bend (SECRBB) test, the Brazilian disc test, and the SCB test under different geometries and crack angles. K. K. Mohammed Shafeeque et al. [24] performed the calibration of double-edge-cracked circular ring specimens. They found that these specimens have the advantage of compressive loading and an enhanced SIF compared to the Brazilian disc specimens. The hole in the center of the disc leads to stress concentration, making these specimens more susceptible to crack extension in the stress concentration region. To date, in most fracture parameter calibrations, only the influence of human controllable parameters such as crack length, crack angle, and crack width on specimen *f* have been considered, while there are few studies considering the influence of petrophysical parameters on specimen *f*. Nejati et al. [25] used an SCB specimen to investigate the influence of crack length, bedding orientation, and material anisotropy on the dimensionless SIFs of an anisotropic material, followed by SIF calibration. However, at present, there is no systematic calibration for HDWS specimens of anisotropic materials.

As a carrier of shale gas, shale is a typical laminated rock with significant anisotropy [26]. Therefore, when calibrating *f* for HDWS specimens, it is not enough to consider only the effect of geometrical parameters on *f* for the specimen; the influence of anisotropic elastic parameters of the material on *f* must also be considered [27]. In actual engineering projects and scientific research, shale is often considered a transversely anisotropic material [28] with a symmetry axis perpendicular to the sedimentary plane [29]. For laminated rock, the bedding orientation is a key factor affecting the test results of SIF and *f*. In three-dimensional space, three typical bedding orientations, namely, arrester (A), divider (D), and short-transverse (ST) [30,31], are often studied. There are five independent elastic parameters for a transversely isotropic material [25,27]: Young’s modulus *E* and Poisson’s ratio *v*′ in the isotropic plane and Young’s modulus *E′*, shear modulus *G′*, and Poisson’s ratio *v′* perpendicular to the isotropic plane. In particular, the ratio of *E* to *E′*, *E/E′* (*ξ*), can be used as a parameter to measure the degree of anisotropy for a transversely isotropic material. The ratio of *G′*_sv_ to *G′*, denoted as *ƞ*, calculated from the Saint-Venant principle 1/*G′*_sv_ = 1/*E* + (1 *+* 2 *v′*)/*E′*, can also be used as a parameter to measure the degree of anisotropy of a transversely isotropic material [25].

As mentioned above, at present, there is no systematic calibration for HDWS specimens of transversely isotropic material. Therefore, it is important to study the fracture of HDWS specimens of transversely isotropic materials to provide data support for the hydraulic fracturing of shale; thus, this paper presents a comprehensive study of the shape factor of HDWS specimens of transversely isotropic materials. This paper begins with a brief description of the HDWS specimen. Then, we used the finite element software ABAQUS 2021 to calculate *f* for the mode I fracture of HDWS specimens containing transversely isotropic material and systematically investigated the influences of crack length, hole size, bedding orientation, Young’s modulus ratio *ξ*, and apparent shear modulus ratio *ƞ* on *f*. The shape factor of the HDWS test was also calibrated using numerical methods. Additionally, a hydraulic fracturing experiment of HDWS specimens on Longmaxi shale in three different bedding orientations (A, D, and ST) was carried out. The fracture toughness is calculated based on the peak pressure and *f*, and the effectiveness of the numerical calibration was verified.

## 2. Numerical Calibration of *f* for Mode I Fracture of HDWS Specimens

### 2.1. HDWS Specimens

As shown in Figure 1, HDWS specimens are long hollow cylindrical tubes, each with a length of *h*, a tube radius of *R*, and a central through-hole radius of *r*. Two cracks, each with a length of *a*, are prefabricated along each specimen on both sides of the central hole symmetrically along the diameter.

The steps for HDWS specimen preparation are as follows: (1) Coring: A cylindrical core in a specific bedding orientation was drilled using a 50-mm drill bit. (2) Cutting: Two cross-sections were cut using a cutting machine. (3) Grinding: The two ends were ground using a grinder until the height of the cylinder was approximately 105 mm. (4) Drilling: The rough-machined specimen was drilled in the center, and the two ends were ground after drilling to make the height of the cylinder within a range of 100 mm ± 0.2 mm. (5) Cutting: The prefabricated cracks were cut into the specimen from the central through-hole using a CNC wire cutting machine.

The cylindrical profile of the HDWS facilitates the simulation of in situ stress by applying confining pressure, and the central through-hole facilitates the testing of rock fracture toughness by hydraulic fracturing. Since hydraulic fracturing is an important part of the shale gas development process, the HDWS specimen is often used to simulate the fracture toughness test of shale at different depths. For isotropic material, *f* of the HDWS specimen is affected by geometric parameters such as the crack length and hole size of the specimen, but for shale, which is a typical transversely anisotropic material, *f* depends on the bedding orientation of the specimen and the anisotropy of the material’s elastic parameters [25]. HDWS specimens with three typical bedding orientations, i.e., A, D, and ST, are shown in Figure 1. In this study, *f* was calibrated for HDWS specimens with these three typical orientations.

### 2.2. Procedure for the Numerical Calibration of f

The SIFs for only certain simple and specific models can be calculated. However, this limitation can be overcome by utilizing finite element models to calculate the SIFs for different and more complex specimen configurations. Therefore, in this paper, the finite element software ABAQUS 2021 was used to calculate the SIFs of HDWS specimens. In essence, three steps are required to complete a finite element method (FEM) analysis: domain discretization, local approximation, and assemblage and solution of the global matrix equation. The domain discretization involves dividing the domain into a finite number of internal contiguous elements of regular shapes defined by a fixed number of nodes. A basic assumption of finite elements is that the unknown function F can be represented by polynomials through the known functions of the individual nodes. This is shown in the following equation:(1)$$u_{i}^{e}=\sum_{j=1}^{M}N_{ij}u_{i}^{j}$$
where the *N_ij_* are often called the shape functions (or interpolation functions). For convenience, Gaussian integrals are used, where *M* is the order of the equation. The original probability density function (PDF) can be expressed as a shape function by the following equation:(2)$$\sum_{i=1}^{N}\left[K_{ij}^{e}\right]\left\{u_{j}^{e}\right\}=\sum_{i=1}^{N}\left(f_{i}^{e}\right)\;\;\;\;\;or\;\;\;\;\;\;\;\;Ku=F$$
where $\left[K_{ij}^{e}\right]$ is the coefficient matrix, $\left\{u_{j}^{e}\right\}$ is the vector of node values of the unknown variables and the vector $\left(f_{i}^{e}\right)$ is controlled by the initial conditions, boundary conditions, and body force terms. For the elasticity problem, the matrix $\left[K_{ij}^{e}\right]$ is called the elemental stiffness matrix, which is given by
(3)$$\left[K_{ij}^{e}\right]=\int_{\Omega_{i}}(\left[B_{i}\right]{\left[N_{i}\right]}^{T})\left[D_{i}\right]\left[B_{j}\right]d\Omega$$
where matrix [*D_i_*] is the elasticity matrix and matrix [[*D_i_*]*_i_*] is the geometry matrix determined by the relation between the displacement and strain. The global stiffness matrix **K** is banded and symmetric because the matrices [*D_i_*] are symmetric. Material inhomogeneity in the FEM is most straightforwardly incorporated by assigning different material properties to different elements (or regions) [32].

Due to the large number of models, three-dimensional calculations are time-consuming, so two-dimensional (2D) models were used in this paper for calculations. As shown in Figure 2, a 2D model of the HDWS specimen was created in ABAQUS. The model has a circular ring shape with a circular hole in the center and prefabricated cracks distributed symmetrically on both sides of the hole. The plane stress assumption was adopted; the height of the model, that is, the thickness of the plane stress, was set to 100 mm, and the ring had an outer radius *R* of 25 mm. To investigate the influence of hole size and crack length on *f* for the HDWS specimen, the inner hole size *r* of the circular ring was set to four levels, i.e., 2.5 mm, 5.0 mm, 7.5 mm, and 10 mm; and the crack length *a* was set to seven levels, i.e., 2.5 mm, 5.0. mm, 7.5 mm, 10.0 mm, 12.5 mm, 15.0 mm, and 17.5 mm. With reference to the elastic parameter tests of the Longmaxi shale by Rui He et al. [33], the elastic parameters of the shale were set as follows: *E* = 24.91 GPa, *v* = 0.166, *v′* = 0.253, and *G* = 6.21 GPa. The physical property parameters of rock vary across different areas or different depths [34,35]. The Young’s modulus ratio *E/E′* (*ξ*) and the apparent shear modulus ratio *G′*_sv_/G′ (*ƞ*) each generally vary within a certain range for most rocks, with *ξ* usually ranging from 1 to 4 and *ƞ* typically ranging from 0.5 to 1.5 [25]. Preliminary research has shown that the change in *ξ* has a pronounced influence on *f*, while the change in *ƞ* has little impact on *f*. Therefore, this study selected HDWS specimens with *ξ* values of 1, 1.5, 2, 2.5, 3, 3.5, and 4 and *ƞ* values of 0.7, 1.0, and 1.3 for investigation. When *ξ* is equal to 1, the specimen is made of an isotropic material; the influence of crack length and hole size on *f* was investigated separately in this isotropic case. The model material parameters *E′* and *G′* were set according to the values of *ξ* and *ƞ*, respectively, to study the effect of the anisotropy of the elastic parameters on *f*.

The meshing and boundary conditions of the HDWS specimen model are shown in Figure 2. The global node size of the model was set to 1 mm. The free division method was used to mesh the specimen except the crack tip using four-node, bilinear, plane stress quadrilateral elements (CPS4). Due to the singularity in the stress field at the crack tip, the first ring of elements around a crack tip was set as quarter-node singular elements, and the rest of the crack tip was meshed into eight rings of quadrilateral elements using the sweeping technique. The horizontal displacements at the top and bottom end nodes of the model were constrained, and the vertical displacements at the left and right end nodes of the model were also constrained. The load applied to the model consists of two parts: an external confining pressure (*P_m_*) to simulate in situ stress and a water injection pressure (*P_in_*) to simulate hydraulic fracturing at the hole and crack surfaces. The absolute magnitudes of these two loads do not affect *f* [36]. Therefore, *P_m_* and *P_in_* were set to 100 MPa each for the subsequent calculation of *f* under these two different load cases.

After the HDWS specimen model was created in ABAQUS 2021, the mode I fracture SIF of the HDWS specimen was calculated by contour integration; then, the mode I fracture *f* for the HDWS specimen can be calculated by Equation (4) [37]:(4)$$f=\frac{K_{I}}{\sigma_{}\sqrt{\pi a}}$$
where *K*_I_ is the numerically calculated mode I SIF, *a* is the length of the single-edge crack, and *σ* is the stress at the crack tip. When the HDWS specimen was used for the mode I fracture test, the load mainly consisted of the confining pressure and the water injection pressure on the hole wall and crack surface. The SIF and *f* for the HDWS specimen under these two types of loads were calculated separately. The mode I fracture *f* was calculated by setting *σ* to the confining pressure (*P_m_*) and the pressure on the hole wall and crack surface (*P_in_*), respectively, under the two loading conditions [36].

According to the handbook on SIFs [37], in the case of an infinite plate, *f* under the two loading conditions was calculated by Equation (5):(5)$$f=1+\left(1-s\right)\left[0.5+0.743{\left(1-s\right)}^{2}\right]$$
where *s* = *a*/(*r* + *a*). To validate the meshing method and the boundary condition setting in this study, a specimen satisfying the conditions of an infinite plate was established; its *f* was calculated numerically and compared with that calculated using the theoretical formula. For an infinite plate, *R*/*r >* 10 and *a*/(*R* − *r*) > 10 [36]. Therefore, the validation specimen was set with an outer radius *R* of 30 mm, an inner radius *r* of 3 mm, and a crack length *a* of 1 mm. *f* was numerically calculated to be 1.7255 under the confining pressure and 1.671 under the hole wall and fracture surface pressure. Under the two loading conditions, *f* calculated by the theoretical formula was 1.6885, which indicates that the differences between the numerical and theoretical values of *f* under the two loading conditions are 2.1% and 1.0%, respectively, fully demonstrating the rationality of the HDWS specimen model, constraints, and load conditions in this study.

### 2.3. Influences of Loading Condition and Geometrical Dimension on f

First, the mode I fracture *f* for the HDWS specimen of isotropic material was calculated, and the influence of crack length and hole size on *f* for the HDWS specimen of isotropic materials was studied. The *f* for the HDWS specimen under *P_m_* or *P_in_* alone was calculated, and the results shown in Figure 3 indicate that the calculated *f* values were nearly identical under the two loading conditions. A comparison of the two cases with ratios of hole size to the outer radii of 0.1 and 0.2 reveals that *f* first decreases and then increases with increasing crack length. However, a comparison of the two cases with ratios of pore size to the outer radii of 0.3 and 0.4 shows that *f* increases with increasing crack length. As the crack length increases, the increase in *f* with crack length accelerates. As the hole size increases, *f* gradually increases, and the difference between *f* values for different hole sizes increases. Many scholars have conducted research on SIF and *f* values with fractured specimens of isotropic materials. Shafeeque et al. [24] investigated a double-edge-cracked circular ring specimen under external diametrical compression as well as its mode I fracture *f*, *F*_1_, and numerically studied the influences of the ratio of inner to outer radius (*r/R*) and the ratio of crack length to outer radius (*a/R*) on *F*_1_. The authors found that for large hole size and crack length, *F*_1_ exhibits a monotonic increase with an increase in crack length or hole size and that the difference between *F*_1_ values for different hole sizes becomes more pronounced as the crack length increases. These results are consistent with the relationship of *f* and crack length for HDWS specimens with large hole sizes, as revealed by the calculation results in our study, further validating the findings in our study. Furthermore, according to the SIF handbook [37], in the case of a finite hole with two finite cracks symmetrically located along the hole diameter in an infinite plane, *f* decreases as the ratio of crack length to hole size increases, which is consistent with the trend of decreasing *f* with increasing crack length for the *r/R* values of 0.1 and 0.2 observed in our study.

### 2.4. Influences of Bedding Angle on f

To study the influence of the bedding angle on *f* for the HDWS specimen of transversely isotropic material, the corresponding *f* was calculated with a Young’s modulus ratio *ξ* of 2 and a shear modulus ratio *ƞ* of 1 for three bedding orientations, i.e., A, D, and ST orientations. As shown in Figure 4, when the crack is long and all other conditions are the same, *f* is the largest in the specimen with an A direction and the smallest in the specimen with an ST direction. The crack length *a* has a significant impact on the difference in *f* between specimens with different bedding orientations. When the crack length is short, the difference in *f* between specimens with different bedding orientations is small (almost the same at *a*/*R* = 0.1). However, as the crack length increases, the difference in *f* between specimens with different bedding orientations gradually increases. Similar findings have also been reported in studies of *f* for other transversely isotropic specimen configurations by other researchers. Nejati et al. [25] examined the SIFs of semicircular fracture specimens with different bedding orientations (A, D, and ST). Their calculation results showed that the *f*, or *Y*_I_, of the semicircular fracture specimen was significantly affected by the bedding orientation. There was a notable difference in the *f*s among the specimens with the three bedding orientations; *Y*_I_ was the largest for the specimen with an A orientation and the smallest for the specimen with an ST orientation, and the difference in *Y*_I_ among the specimens with the three bedding orientations increased significantly with increasing crack length.

### 2.5. Influences of ξ and ƞ on f

For the specimen with a D orientation, both the bedding planes and their normals are perpendicular to the crack plane. In the 2D case, the calculation of *f* can be simplified to an isotropic problem and is not affected by the Young’s modulus ratio *ξ* or the apparent shear modulus ratio *ƞ*. For this reason, the influence of *ξ* and *ƞ* on *f* for the specimen with a D orientation is no longer considered in the calculation. However, there is a pronounced difference between the influences of *ξ* and *ƞ* on *f*s for specimens with A and ST orientations, respectively. Therefore, the *f*s for HDWS specimens with A and ST orientations were investigated considering different values of the Young’s modulus ratio *ξ* and apparent shear modulus ratio *ƞ*. The *f* values corresponding to *ξ* values of 1.5, 2, 2.5, 3, 3.5, and 4 and *ƞ* values of 0.7, 1.0, and 1.3 were calculated using ABAQUS 2021 software, and the results are shown in Figure 5, Figure 6, Figure 7 and Figure 8.

For specimens with an A orientation, the difference in *f* between specimens with different *ξ* is small when the crack length is small, and such a difference in *f* gradually increases with increasing crack length. For specimens with an A orientation and an *r/R* of 0.1 and 0.2, *f* decreases with increasing *ξ* when the crack length is small, while in other cases, *f* increases with increasing ξ. Consistent with the case of specimens with an A orientation, the difference in *f* between specimens with an ST orientation and different *ξ* is small when the crack is short, and such a difference in *f* gradually increases as the crack grows. Different from the case of specimens with an A orientation, for specimens with an ST orientation and an *r/R* of 0.1 and 0.2, when the crack is short, *f* increases with increasing ξ, while in other cases, *f* shows an increasing trend with increasing ξ. With other conditions being the same, the influence of *ξ* on *f* is less for specimens with an ST orientation than for specimens with an A orientation. The influence of *ƞ* on *f* is less than the influence of *ξ* on *f*. For specimens with an A orientation, *f* increases with increasing *ƞ*; for specimens with an ST orientation, *ƞ* has almost no influence on *f*, with the difference in *f* under different *ƞ* values being less than 1%. Similar findings have been reported by other scholars. Nejati [25] studied the influence of Young’s modulus ratio *E/E′* and the apparent shear modulus ratio *G′*_sv_/G′ on the modified dimensionless SIF, or *Y*_I_, of a semicircular specimen. Their test results showed that the *Y*_I_ of a specimen with an A orientation similarly increases with the increase in *E/E′*, that the *Y*_I_ of the specimen with an ST orientation decreases with the increase in *E/E′*, and that *G’*_sv_*/G′* has little impact on *Y*_I_ for specimens with A and ST orientations, which is in basic agreement with our *f* calculation results for HDWS specimens with the two bedding orientations, thereby verifying the correctness of the results in this study.

## 3. Hydraulic Fracturing Experiment of HDWS Specimens

The specimens tested in the experiment were collected from the Longmaxi shale outcrop in Chongqing, China, and have a density of 2.66 g/cm^3^. Through a series of uniaxial compression experiments, the five elastic parameters, *E*, *E′*, *G′*, *v*, and *v′*, were measured to be 24.91 GPa, 20.87 GPa, 10.68 GPa, 0.166, and 0.212, respectively [33]. The shale was processed into HDWS specimens, each with an outer radius *R* of 25 mm, a hole radius *r* of 2.5 mm, prefabricated cracks of length *a* of 4 mm on both sides, and a height *h* of 100 mm. Three typical bedding orientations, i.e., the A, D, and ST orientations, were considered, for each of which two specimens were prepared, leading to a total of six specimens. Due to the small number of rock samples available for the experiment, only six specimens were tested experimentally, and it was not possible to compare the experimental conditions of shale from different regions.

The hydraulic fracturing experiments on the HDWS specimens of the Longmaxi shale with three different bedding orientations (A, D, and ST) were carried out using the multiphysics coupling triaxial testing system (Top Industrie, France) at Sichuan University. The triaxial chamber of the apparatus can measure specimens with a standard radius of 50 mm or 25 mm and has a maximum axial load of 2000 kN, a maximum confining pressure of 100 MPa, linear variable differential transformers (LVDTs) with a range of 100 mm for measuring the machine displacement, LVDTs with a range of ±50 mm and ±10 mm for measuring the longitudinal deformation of the specimen, a maximum liquid osmotic pressure of 100 MPa, and a maximum gas osmotic pressure of 100 MPa. The HDWS specimen loading mode is shown in Figure 9. This loading mode can simulate the in situ stress and water injection pressure in hydraulic fracturing. This experiment was conducted only to verify the *f* obtained from the numerical simulation, so it was performed under a confining pressure of 0 MPa. The specimen was connected to the sealing heads at the top and bottom, with O-rings installed inside the sealing heads to effectively prevent the leakage of water and the pressure within the hole after the application of axial pressure. In the experiment, the axial pressure was applied at a rate of 0.2 MPa/s to a target stress of 5 MPa, and then the stress state was held constant for 2 min. After the specimen was fixed, distilled water was injected as a fracturing fluid into the specimen from its bottom at a constant rate of 4 mL/min using a servo-controlled injection system. After the specimen was fractured, the fracturing fluid continued to be injected for a period to stabilize the pressure inside the triaxial chamber, thus bringing the entire experimental system to an equilibrium state.

The water injection pressure-time curves of the six specimens are shown in Figure 10. The experimental results exhibit a consistent trend with typical pressure-time curves obtained in hydraulic fracturing tests by Sarmadivaleh et al. [38] A typical pressure-time curve is divided into four main stages: an initial pressure development stage, a pressurization stage, a fracturing stage, and a post-failure stage. In the initial pressure development stage, the curve grows slowly. During the pressurization stage, the curves of different specimens show a relatively consistent upward trend, with the water injection pressure increasing linearly until reaching the fracturing stage. Due to the release of a certain pressure after the failure of the specimen induced by hydraulic fracturing, a noticeable pressure drop can be observed in the fracturing stage, and the residual pressure of the specimen is almost zero after fracturing. The fracturing pressure curves of the HDWS specimens with different bedding orientations generally exhibit consistent trends, but the peak pressure exhibits anisotropy. In the experiment, the peak pressures are 8.61 MPa and 11.11 MPa for HDWS specimens with an A orientation, 8.67 MPa and 9.29 MPa for specimens with a D orientation, and 6.18 MPa and 7.70 MPa for specimens with an ST orientation. According to the experimental results, specimens with an ST orientation have the minimum peak pressure; specimens with A and D orientations have similar peak pressures, but specimens with an A orientation have a higher average peak pressure than specimens with a D orientation.

With the method described earlier, the *f*s for the HDWS specimens in the experiment are calculated to be 1.39217, 1.38653, and 1.38320 for specimens with A, D, and ST orientations, respectively. The fracture toughness of shale in each bedding orientation can be obtained by
(6)$$K_{Ic}=P_{b}\sqrt{\pi a}f$$
where *P_b_* is the peak pressure obtained in the experiment and *f* is the shape factor of the specimen. The average fracture toughness is calculated to be 1.54 MPa·m^1/2^ for specimens with an A orientation, 1.40 MPa·m^1/2^ for specimens with a D orientation, and 1.08 MPa·m^1/2^ for specimens with an ST orientation. Hence, shale exhibits a certain degree of anisotropy in fracture toughness, with the highest and lowest fracture toughnesses observed in specimens with A and ST orientations, respectively. Similar results have been reported in experimental studies by other scholars. Heng et al. [39] tested the fracture toughness of shale by using cylindrical specimens with straight-cut notches and the three-point bending method and obtained an average fracture toughness of 1.146 MPa·m^1/2^, 0.957 MPa·m^1/2^, and 0.566 MPa·m^1/2^ for specimens with A, D, and ST orientations, respectively. These results indicate that shale bedding planes have a low resistance to crack propagation and that cracks tend to propagate and extend along the bedding planes in specimens with an ST orientation, while in specimens with an A orientation, the cracks are perpendicular to the bedding planes and encounter the highest resistance during propagation and extension.

## 4. Conclusions

In this study, the influences of crack length, hole size, and elastic parameter anisotropy on the *f* of specimens with three different bedding orientations were systematically investigated. The numerical calibration was verified by hydraulic fracturing experiments using HDWS specimens from the Longmaxi shale and fracture toughness calculations in conjunction with *f*.

(1)Numerical simulation results show that the mode I *f* of the HDWS specimen increases monotonically with the increase in hole size. The pattern of variation in *f* with crack length is affected by the size of the hole. Furthermore, as the crack length increases, *f* increases at an accelerated rate.(2)There is clear anisotropy in *f*. With all other parameters being the same, *f* is the largest for specimens with an A orientation and the smallest for specimens with an ST orientation. A larger hole and a longer crack lead to more significant anisotropy in *f*.(3)The degree of anisotropy in elastic parameters has an important impact on *f*. A higher Young’s modulus ratio *E/E′* results in a larger *f* for specimens with an A orientation, while it leads to a smaller *f* for specimens with an ST orientation. The apparent shear modulus ratio *G′*_sv_/*G′* has relatively little impact on *f*.(4)The results from the hydraulic fracturing experiments indicate anisotropy in the peak pressure of specimens with different bedding orientations, with the peak pressure and fracture toughness of the samples in the ST direction being the lowest and those in the A direction being the highest.

## Figures and Tables

**Figure 1 materials-16-06754-f001:**
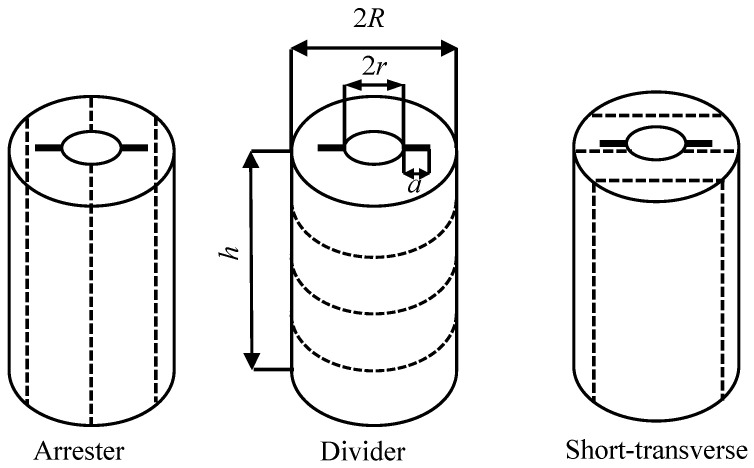
HDWS specimens in three typical bedding orientations.

**Figure 2 materials-16-06754-f002:**
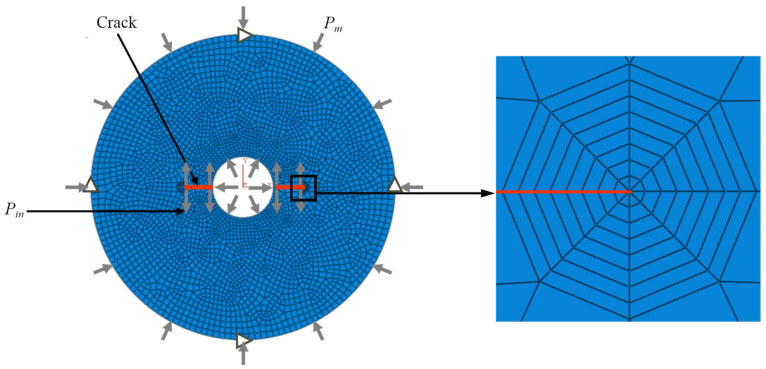
The overall mesh and a close-up of the crack tip mesh of the 2D model for the HDWS specimen.

**Figure 3 materials-16-06754-f003:**
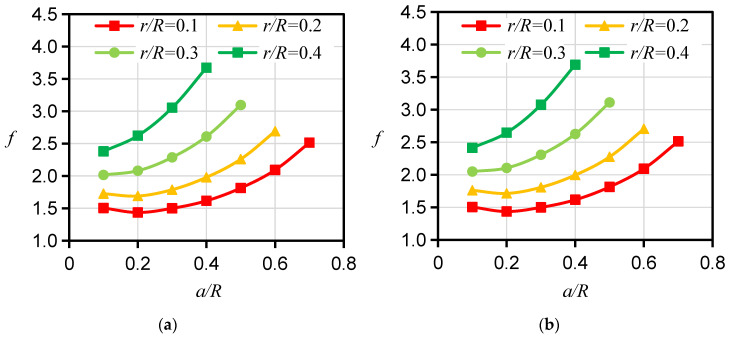
Calculation results of *f* for the HDWS specimen of isotropic material with different *a*/*R* values under *P_in_* or *P_m_* loading. (**a**) *f* under *P_in_* loading. (**b**) *f* under *P_m_* loading.

**Figure 4 materials-16-06754-f004:**
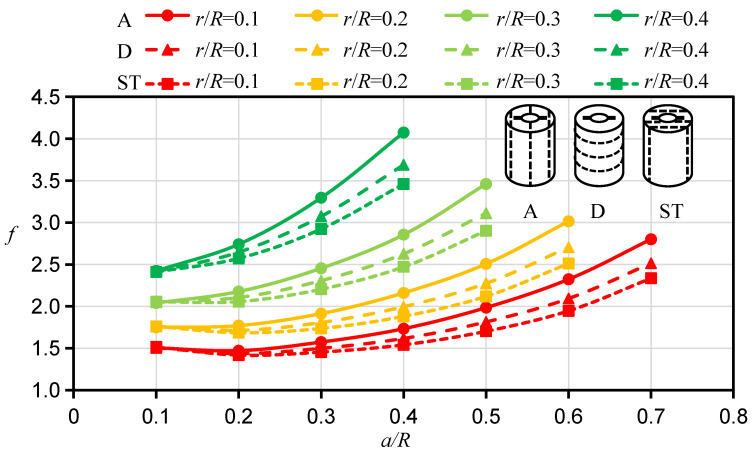
Calculation results of *f* for HDWS specimens with three bedding orientations.

**Figure 5 materials-16-06754-f005:**
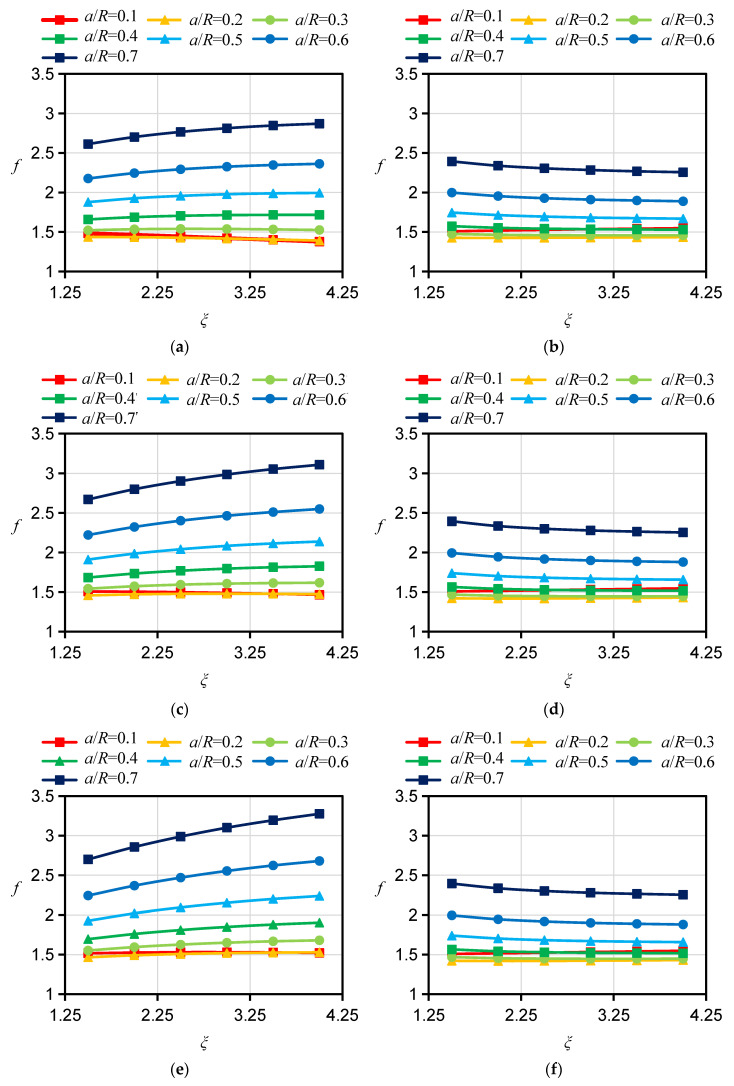
Results of *f* for specimens with *r*/R = 0.1 and different material anisotropies. (**a**) A orientation *ƞ* = 0.7. (**b**) ST orientation *ƞ* = 0.7. (**c**) A orientation *ƞ* = 1.0. (**d**) ST orientation *ƞ* = 1.0. (**e**) A orientation *ƞ* = 1.3. (**f**) ST orientation *ƞ* = 1.3.

**Figure 6 materials-16-06754-f006:**
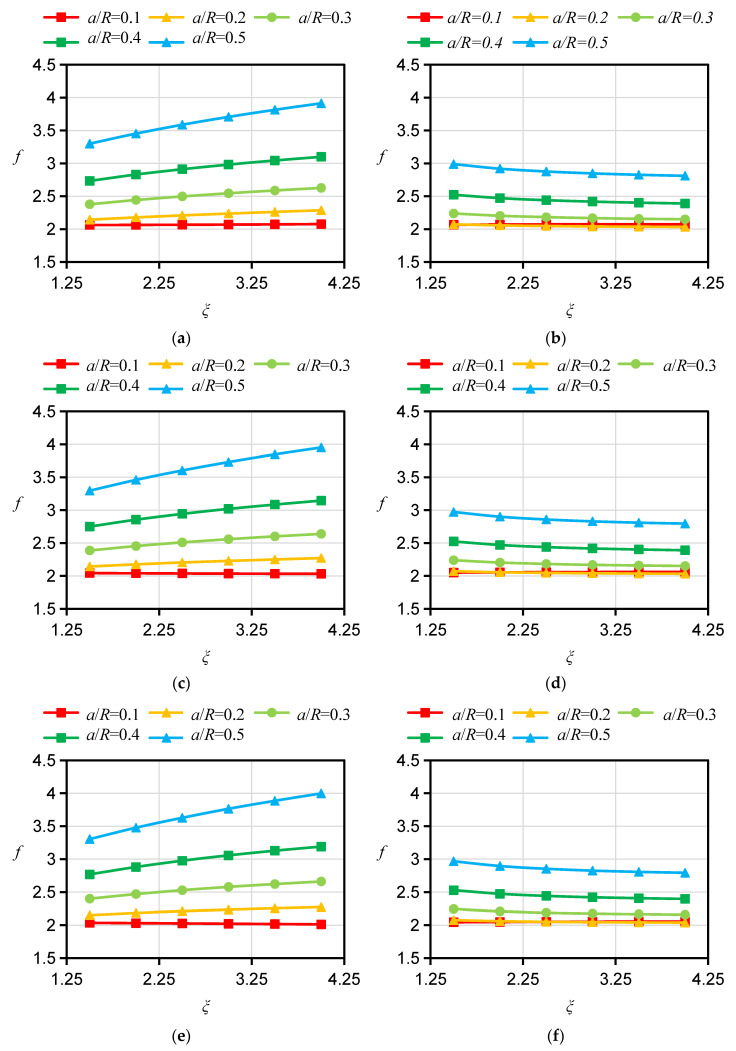
Results of *f* for specimens with *r*/R = 0.3 and different material anisotropies. (**a**) A orientation *ƞ* = 0.7. (**b**) ST orientation *ƞ* = 0.7. (**c**) A orientation *ƞ* = 1.0. (**d**) ST orientation *ƞ* = 1.0. (**e**) A orientation *ƞ* = 1.3. (**f**) ST orientation *ƞ* = 1.3.

**Figure 7 materials-16-06754-f007:**
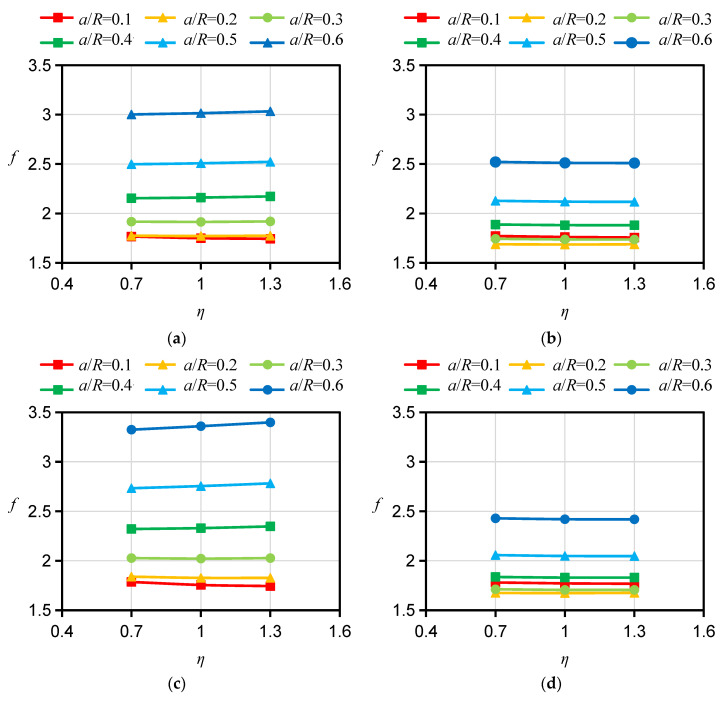
Results of *f* for specimens with *r*/R = 0.2 and different material anisotropies. (**a**) A orientation *ξ* = 2.0. (**b**) ST orientation *ξ* = 2.0. (**c**) A orientation *ξ* = 3.5. (**d**) ST orientation *ξ* = 3.5.

**Figure 8 materials-16-06754-f008:**
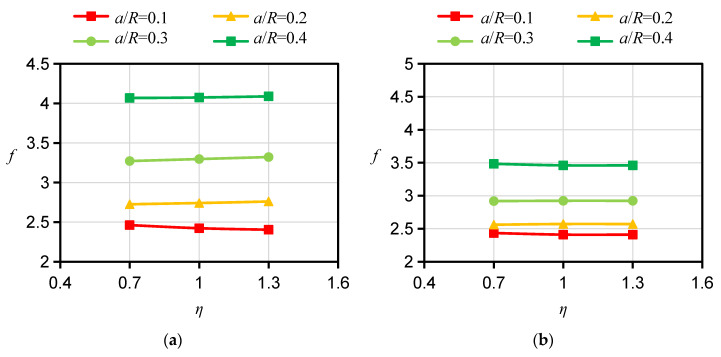
Results of *f* for specimens with *r*/R = 0.4 and different material anisotropies. (**a**) A orientation *ξ* = 2.0. (**b**) ST orientation *ξ* = 2.0. (**c**) A orientation *ξ* = 3.5. (**d**) ST orientation *ξ* = 3.5.

**Figure 9 materials-16-06754-f009:**
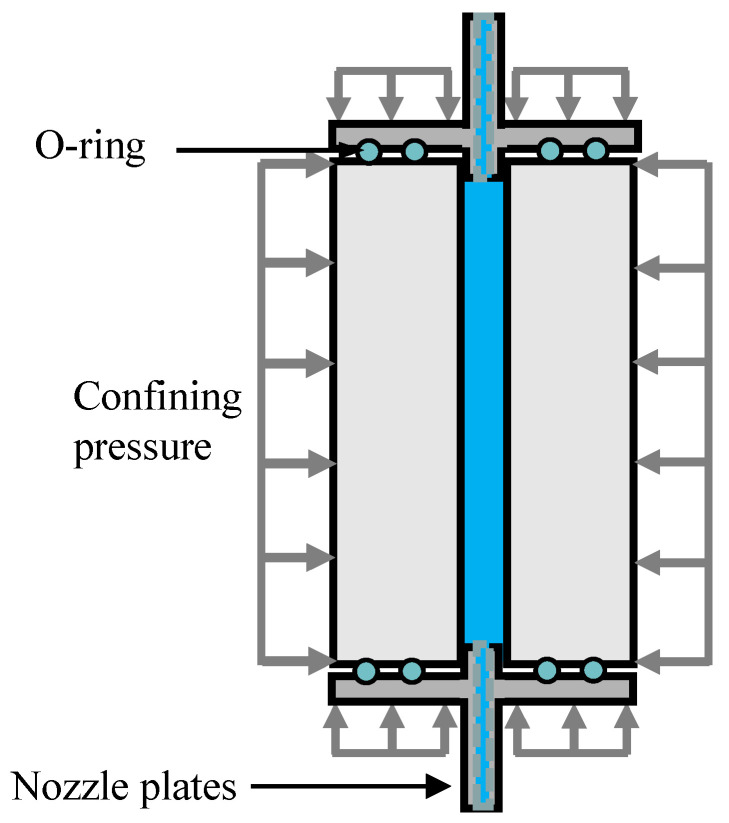
Schematic diagram of loading of an HDWS specimen in the fracture test.

**Figure 10 materials-16-06754-f010:**
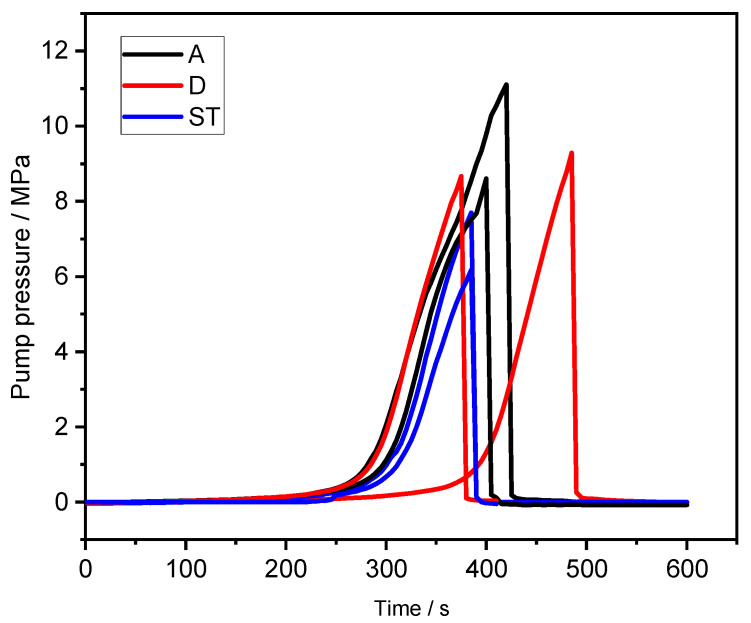
Fracture pressure-time curves of specimens with different bedding orientations (two specimens for each bedding orientation).

## Data Availability

No new data were created or analyzed in this study. Data sharing is not applicable to this article.

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
