# Peer review of "Fracture Toughness Testing of Brittle Laminated Geomaterials Using Hollow Double-Wing Slotted Specimens"

_materials, 2023, doi:10.3390/ma16206754_

Round 1

Reviewer 1 Report

This paper presents the fracture toughness testing of brittle laminated geomaterials using hollow double-wing slot specimens. The following points need to be clarified:

  1. The abstract is too long, it needs to be shortened. Only the main points need to be mentioned. Also, the parameters such as E should be italics. Implement this for all the other parameters.
  2. More updated research results could be added in the literature review.

3.      It is suggested that the authors should modify the last part of the introduction to: 1) clearly mention the goal and novelty of this work, 2) mention the methodology used and its important to place the major hypothesis, 3) mention the structure of the paper or procedure of their work.

4.      Figs 4 to 9 presents lots of results all together. It is recommended to divide this part into subsections and analyze the influence of each parameter separately.

5.      The authors should explain the importance of the developed FE model. The obtained results can be also obtained via theoretical model. Is not it?

6.      Identify or show the values found in Figs 4 to 9 inside the figure with some characters such as using ., *, +).

7.      The paper lacks a discussion on the weakness and limitations of the present methodology proposed.

8.      There exist some typos: Page 13: Reference source not found. Page 8: subsections e, f, c and. The proofreading of the paper needs to be done.

  1. The conclusion is too long. Only the key findings should be highlighted.

Moderate editing of English language required

Reviewer 2 Report

Review report. Major corrections

In the manuscript "Fracture toughness testing of brittle laminated geomaterials using hollow double-wing slot specimens", the f for HDWS specimens of transversely isotropic material is calibrated using the finite element software ABAQUS, and the influences of crack length, hole size, and the anisotropy of elastic parameters on f under three typical bedding orientations—arrester (A), divider (D), and short-transverse (ST)—were systematically investigated.

Comments on the overall concept: The paper is interesting. The results are applicable in engineering practice. In the terms of science, the topic is relatively novel. According to employed the anti-plagiarism engine, this manuscript is not published elsewhere.  

Specific comments:

1. The abstract is too long. It defines and describes the procedure employed in the research instead of just highlighting the concept of the work, scientific novel and some of the main results. it should be corrected accordingly.

2. The title and keywords are informative.

3. Introduction comprises relatively good review of the literature. The text is understandable and easy to follow. However, there are a lot of typing mistakes, as well as formatting mistakes.

4. Figure 1 is unnecessary.

5. Please, carefully reread and correct entire chapter 2. Numerical calibration of f for mode-I fracture of HDWS specimens. There are numerous formatting mistakes which make the text unreadable.

6. Since there is no numeration of lines, I cannot point out on the exact line, but in Page 4, what is going on with Figure 2. Is it one figure or two figures. Each figure should be positioned and quoted in the text. The same situation is on the page 5. Please correct all of the figures and place them where they should be.

7. the quality of figures is questionable. it has to be improved.

8. Figures on the page 9 and forward – can results be on the same diagram? Or at least a reasonable number of diagrams.

9. Formatting on the page 13 is completely off.

10. Conclusions are highlighting the main findings in the research.

11. The listed references are relevant and up-to-date (within the last 5 years or so). The used literature is appropriate. There are no excessive self-citations.

A native English speaker or a professional proofreader should read and revise the text.

Reviewer 3 Report

The authors have studied the fracture toughness of brittle laminated geomaterials using hollow double-wing slot specimens with the help of ABAQUS. The area of the paper is interesting. However, some suggestions/queries are given below that need to be answered.

The abstract should be focused on the find and its values. The reasoning may be discussed in the discussion section. Overall, the abstract needs to be shortened.

There are many typos and formatting errors that need to be checked and corrected in the updated version.

Figure 2 is repeated thrice why?

What is the version of the software ABAQUS? Kindly mention this in the paper.

The figures are not cited in the body of the paper. Kindly cite the figure wherever it has been discussed.

Do the results are verified for different mesh sizes to have grid sensitivity analysis?

The authors have not discussed the governing equations and different schemes used for the simulation. Kindly include the physics involved in the calculation. It will enhance the quality of the paper and improve the readability of the paper.

Where are the explanations for figure 6 to figure 9? Similarly, for other figures.

Okay but needs minor revision.

Round 2

Reviewer 1 Report

The paper is revised adequately.

Ok

Reviewer 2 Report

The manuscript is corrected enough to merit the publication.

Reviewer 3 Report

The authors have made the required corrections.